# Suppressing photochemical reactions with quantized light fields

Javier Galego[1], Francisco J. Garcia-Vidal[1,2] & Johannes Feist[1]

Photoisomerization, that is, a photochemical reaction leading to a change of molecular structure after absorption of a photon, can have detrimental effects such as leading to DNA damage under solar irradiation, or as a limiting factor for the efficiency of solar cells. Here, we show that strong coupling of organic molecules to a confined light mode can be used to strongly suppress photoisomerization, as well as other photochemical reactions, and thus convert molecules that normally show fast photodegradation into photostable forms. We find this to be especially efficient in the case of collective strong coupling, where the distribution of a single excitation over many molecules and the light mode leads to a collective protection effect that almost completely suppresses the photochemical reaction.

[1] Departamento de Física Teórica de la Materia Condensada and Condensed Matter Physics Center (IFIMAC), Universidad Autónoma de Madrid, Madrid E-28049, Spain. [2] Donostia International Physics Center (DIPC), Donostia/San Sebastian E-20018, Spain. Correspondence and requests for materials should be addressed to F.J.G.-V (email: fj.garcia@uam.es) or to J.F. (email: johannes.feist@uam.es).

Photoisomerization is one of the most fundamental photochemical processes. It can perform desirable functionality, for example, as the primary photochemical event in human vision, where it stores electronic energy in the molecular structure[1,2], or for possible applications in solar energy storage[3] and as memories, switches and actuators[4,5]; but it can can also have detrimental effects, for example as an important damage pathway under solar irradiation of DNA[6,7], or as a limiting factor for the efficiency of organic solar cells[8]. While photoisomerization can be avoided by shielding the system from light, this is of course not a viable pathway for approaches that rely on the interaction with external light (such as solar cells or solar energy storage). For these applications, it would therefore be desirable to control or suppress photochemical reactions while still allowing or even enhancing the interaction with external light. As we show below, this could be achieved by exploiting strong coupling with confined light modes.

Strong coupling occurs when the coherent energy exchange between molecules and a light mode becomes faster than the decoherence processes in the system[9,10]. This creates paradigmatic hybrid quantum systems with eigenstates that have mixed light-matter character (so-called polaritons). Organic materials provide particularly large dipole moments and high molecular densities, making them ideal systems to reach the strong coupling regime[11,12]. By exploiting the strong field localization in plasmonic nanocavities, even single-molecule strong coupling has recently been achieved[13]. In addition to demonstrating strong coupling, pioneering experiments have shown modifications of material properties under strong coupling[14–17]. In particular, Hutchison *et al.*[14] showed that the rate of a photochemical reaction (photoisomerization from spiropyran to merocyanine) can be modified. At the same time, most theoretical descriptions of strong coupling are based on two-level systems, which cannot address such effects in molecules with many nuclear (that is, rovibrational) degrees of freedom. First theoretical treatments of the influence of strong coupling on internal degrees of freedom have only appeared recently[18–22].

We here demonstrate that a wide class of photochemical reactions can be strongly suppressed under strong coupling. In this regime, the hybrid light-matter potential energy surfaces (PES) of the molecules develop new minima in which the excited-state wavepackets are trapped after excitation. Furthermore, we show that this effect is more pronounced when many molecules are coupled to the light mode due to a 'collective protection' effect. Our results imply that even very fragile molecules could be stabilized by simply putting them close to a nanophotonic structure.

## Results

**Single-molecule strong coupling.** We treat a general molecular model that can represent a variety of commonly studied photoisomerization reactions, such as *cis-trans* isomerization of stilbene, azobenzene or rhodopsin[2,23,24] (corresponding to rotation around a C=C or N=N double bond, as sketched in the inset of Fig. 1), or ring-opening and ring-closing reactions in diarylethenes[4]. The model molecule (see Methods for more detail) describes nuclear motion on ground and excited electronic PES along a single reaction coordinate $q$, as shown in Fig. 2a. All other degrees of freedom are assumed to be fully relaxed, such that the excited PES represents the minimum-energy reaction path. The ground state PES, $V_g(q)$ (blue line), possesses minima at $q = q_0 \approx -1.05$ a.u. and $q \approx 1$ a.u., corresponding to the stable (for example, *trans*-) and metastable (for example, *cis*-) isomers, respectively. They are separated by a barrier with a maximum at $q \approx 0$ accompanied by an avoided crossing between

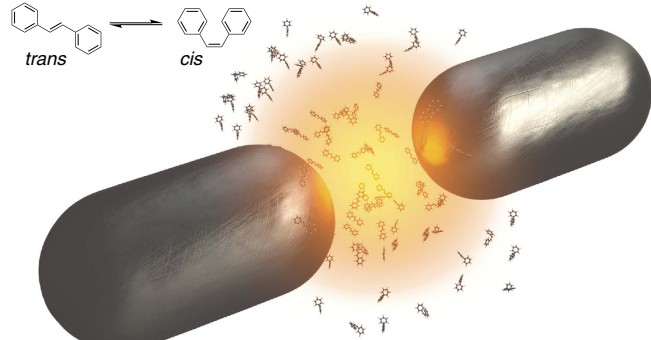

**Figure 1 | Sketch of an example system.** A collection of molecules is coupled to a localized surface plasmon mode in the gap between two nanoparticles. Inset: Sketch of photoisomerization reaction between a *trans*- and *cis*-isomer.

the ground and excited state PES, $V_e(q)$ (orange line). In order to ensure a large quantum yield for photoisomerization in the bare molecule, we choose a very narrow avoided crossing (with energy splitting 39 meV, smaller than the width of the lines in Fig. 2). A wavepacket that encounters the avoided crossing then undergoes an efficient nonadiabatic transition (that is, follows the diabatic surfaces, see Methods), as shown in Fig. 2d using full wavepacket propagation after excitation from the ground to the excited electronic state by an ultrashort laser pulse. For the chosen parameters, the bare model molecule undergoes rapid photoisomerization, with the nuclear wavepacket reaching the second isomer ($q > 0$) within a few hundred fs.

In contrast, when the system enters strong coupling, photoisomerization in a single molecule is suppressed. To show this, we rely on the theoretical framework we introduced in ref. 19, which extends the well-known Born-Oppenheimer approximation with the tools of cavity quantum electrodynamics (QED) by including the light-matter interaction in the 'electronic' Hamiltonian and following nuclear dynamics on hybrid light-matter PES. We include a (single) quantized light mode (which can represent confined light modes in different physical systems, such as microcavity modes or localized surface plasmon resonances as in Fig. 1) with energy term $\omega_c \hat{a}^\dagger \hat{a}$. Here, $\omega_c$ is the quantized mode frequency, and $\hat{a}^\dagger$ and $\hat{a}$ are the associated bosonic creation and annihilation operators. The light-matter coupling is given by $\hat{\mu}(q) \cdot \mathbf{E}_{1ph}(\hat{a}^\dagger + \hat{a})$, where $\mathbf{E}_{1ph}$ is the electric field amplitude of a single quantized confined photon, and $\hat{\mu}$ is the (vectorial) molecular dipole operator. Without light-matter coupling, the photonically excited surface describes the motion of a ground-state molecule with an (uncoupled) photon present in the cavity, and is thus simply a copy of the molecular ground state shifted upwards by the photon energy, $V_c(q) = V_g(q) + \omega_c$ (purple curve in Fig. 2a). When coupling is turned on, the two singly excited surfaces $V_c(q)$ and $V_e(q)$ hybridize, forming 'polaritonic' surfaces with mixed light-matter character, as depicted in Fig. 2b,c. The splitting between the polaritonic PES around equilibrium ($q_0 \approx -1.05$ a.u.) is approximately equal to the Rabi frequency $\Omega_R = 2\mu_{eg}(q_0) \cdot \mathbf{E}_{1ph}$. Importantly, the lower polariton PES develops a deeper and deeper minimum as the coupling is increased. This has two primary reasons: First, the light-matter coupling is most effective when $V_c(q)$ and $V_e(q)$ are close, 'pushing down' the lower polariton. At regions of larger detuning, the 'polaritonic' PES are almost identical to the uncoupled ones. Second, the local shape of the polariton PES becomes a mixture of the two uncoupled PES in regions where they hybridize significantly. Since the photonic surface $V_c(q)$ behaves like the ground-state PES, this additionally supports the formation of a local minimum

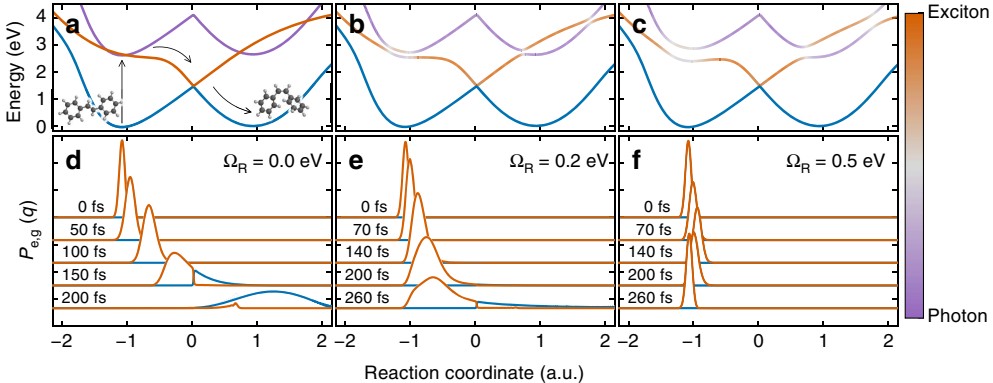

**Figure 2 | Suppression of photoisomerization under strong coupling for a single molecule.** (**a–c**) Ground (blue) and excited (purple-orange colour scale) potential energy surfaces of the model molecule coupled to a quantized light mode ($\omega_c = 2.65$ eV), with the light-matter coupling strength $\Omega_R$ increasing from **a–c**. The continuous colour scale encodes the nature of the hybridized excited PES. (**d–f**) Time propagation of the nuclear wavepacket after sudden excitation to the lowest excited PES ($=$ lower polariton for $\Omega_R > 0$), shown separately for the parts in the lower polariton surface (orange) and the ground state surface (blue) reached through the nonadiabatic transition at $q = 0$. Contributions in the upper polariton surface are negligible and not shown.

in the polaritonic PES. In combination, this leads to the formation of a reaction barrier against isomerization as the coupling is increased, as seen in Fig. 2b,c. At intermediate coupling, where no barrier is formed yet, the reaction is slowed down, but not suppressed (see Fig. 2b,e). Once the coupling becomes sufficiently large, a barrier appears and the excited wavepacket is trapped in the local minimum, such that isomerization becomes almost completely suppressed (see Fig. 2c,f). The initial wavepacket on the lower polariton surface in our calculations is started by a sudden transition and thus includes all vibrationally excited states that are reachable from the ground state through a dipole transition. If the coherent excited wavepacket is successfully trapped without undergoing ultrafast isomerization (as in Fig. 2c,f), the ultimate fate of the molecule will be determined by two additional effects: On the one hand, the excited wavepacket will thermalize within the excited polariton PES on typical timescales of picoseconds. While the exact values depend on the details of the system, we note that for the model molecule treated here, the barrier height of $\approx 65$ meV in Fig. 2c is much larger than the thermal energy $k_B T \approx 26$ meV at room temperature, preventing isomerization. On the other hand, the excited-state wavepacket will simultaneously decay both by radiative and nonradiative processes with timescales typically dominated by the photonic part of the polaritonic PES, ranging from tens of femtoseconds for plasmonic resonances to picoseconds and longer for dielectric structures.

Note that while the upper polariton PES appears even more stable than the lower one in this model, this is an artefact of the restriction to one degree of freedom, with all other degrees of freedom relaxed to their local minimum. This implies that the lower polariton PES indeed corresponds to the lowest-energy excited state, such that the restriction to one coordinate is well-justified. In contrast, the upper polariton surface can possess efficient relaxation pathways to the lower polariton along orthogonal degrees of freedom, and indeed, upper polaritons are known to decay relatively quickly within the excited-state subspace[25,26].

We have thus shown that strong coupling of a single molecule to a confined light mode can strongly suppress photoisomerization reactions and stabilize the molecule. The recent experimental realization of single-molecule strong coupling proves that this could indeed be a viable pathway towards manipulation of single molecules[13]. At the same time, most experiments achieving strong coupling with organic molecules have exploited *collective* coupling[27,28], in which $N \gg 1$ molecules coherently interact with a single mode, leading to an enhancement of the total Rabi

frequency by a factor of $\sqrt{N}$ (ref. 29). In this context, it should be noted that since the single-photon electric field strength decreases with the effective mode volume of the EM mode, $|\mathbf{E}_{1ph}| \propto V^{-1/2}$, the achievable Rabi splitting $\Omega_R \propto \sqrt{N/V}$ depends on the molecular density $\rho \propto N/V$ instead of the absolute number of molecules[30]. This explains why the Rabi splittings achieved in organic systems are within an order of magnitude ($\approx 0.1$–1 eV) for EM modes with vastly different mode volumes.

However, it has recently been shown that in contrast to the Rabi splitting, many observables corresponding to 'internal' degrees of freedom of the molecules are only affected by the single-molecule coupling strength and thus not strongly modified under collective strong coupling[19,20]. One could thus expect that the suppression of photoisomerization disappears when $N$ is sufficiently large. We next show that exactly the opposite is the case, and strong coupling of a large number of molecules to a single mode actually improves the molecular stabilization significantly.

**Collective strong coupling.** In order to treat collective strong coupling involving $N$ molecules and a single confined light mode, we again restrict ourselves to the zero- and single-excitation subspace. The molecules now have $N$ total nuclear degrees of freedom, described by the vector $\mathbf{q} = (q_1, \ldots, q_N)$, and the PES accordingly become $N$-dimensional surfaces. For the uncoupled system, these surfaces are the global ground state $V_G(\mathbf{q}) = \sum_i V_g(q_i)$, the photonically excited state $V_C(\mathbf{q}) = V_G(\mathbf{q}) + \omega_c$, and the $N$ molecular excited states $V_E^{(i)}(\mathbf{q}) = V_G(\mathbf{q}) + V_e(q_i) - V_g(q_i)$. The electronic-photonic Hamiltonian in the first excited subspace is then given by

$$\hat{H}(\mathbf{q}) = \begin{pmatrix} V_C(\mathbf{q}) & g(q_1) & g(q_2) & \cdots & g(q_N) \\ g(q_1) & V_E^{(1)}(\mathbf{q}) & 0 & \cdots & 0 \\ g(q_2) & 0 & V_E^{(2)}(\mathbf{q}) & \cdots & 0 \\ \vdots & \vdots & \vdots & \ddots & \vdots \\ g(q_N) & 0 & 0 & \cdots & V_E^{(N)}(\mathbf{q}) \end{pmatrix}, \quad (1)$$

where $g(q) = \boldsymbol{\mu}_{eg}(q) \cdot \mathbf{E}_{1ph}$. Diagonalizing $\hat{H}(\mathbf{q})$ gives $N+1$ polaritonic surfaces, which describe the collective coupled motion of all molecules. In principle, this could induce, for example, collective transitions in which multiple molecules move in concert. We show in Fig. 3a that this is not the case. Here, we plot the lower-polariton PES (the lowest excited-state surface) as a function of the reaction coordinates of the first two molecules, $q_1$ and $q_2$, while keeping all other molecules fixed to the equilibrium

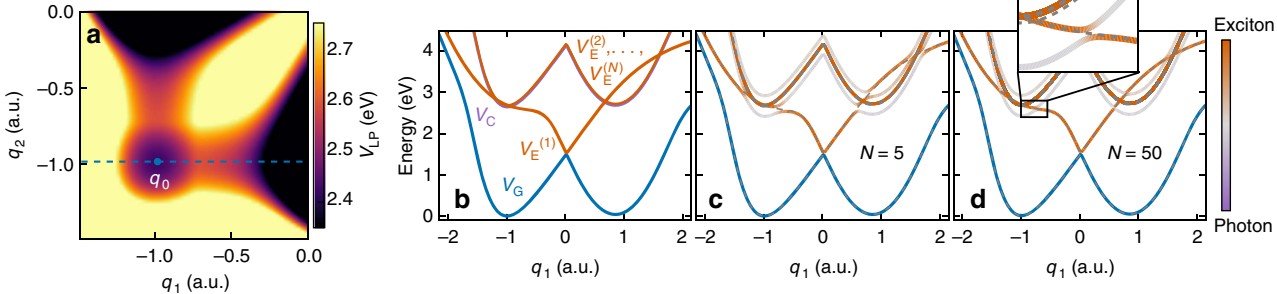

**Figure 3 | Many-molecule potential energy surfaces under strong coupling.** (**a**) Lower polariton PES for $N = 50$ molecules, under motion of molecules 1 and 2, with all others held in the equilibrium position $q_0$. (**b–d**) All potential energy surfaces under motion of only molecule 1, for no light-matter coupling (**b**), and under strong coupling for $N = 5$ (**c**) and $N = 50$ (**d**) molecules. In all panels, the photonic mode frequency is $\omega_c = 2.65$ eV, while the (collective) Rabi frequency is fixed to $\Omega_C = \sqrt{N}\Omega_R = 0.5$ eV.

position ($q_j = q_0$ for $j > 2$). The figure clearly shows that the smallest barrier for isomerization starting from the ground-state equilibrium position $\mathbf{q}_0 = (q_0, \ldots, q_0)$ occurs for motion along only one molecular degree of freedom. This can be understood from the fact that each of the $N + 1$ uncoupled PES that combine to form the lower polariton describe the nuclear motion of $N$ (for $V_C$) or $N - 1$ (for $V_E^{(i)}$) molecules in the ground state. The simultaneous motion of more than one molecule thus necessarily corresponds to motion in the ground-state potential wells, that is, along steep potential barriers. To verify this, we have explicitly checked that the barrier for isomerization rapidly increases with $n$ for simultaneous motion of $n$ molecules ($q_j = q$ for $j \leq n$, and $q_j = q_0$ for $j > n$).

We thus analyse the coupled states under motion of only the first molecule $q_1$, fixing all other molecules to the ground-state equilibrium position ($q_j = q_0$ for $j > 1$). The corresponding PES are shown in Fig. 3b–d. When the light-matter coupling is zero (Fig. 3b), the surface $V_E^{(1)}(\mathbf{q})$ behaves like $V_e(q_1)$, while all other surfaces (corresponding to photonic excitation, or excitation of a 'stationary' molecule $j > 1$) appear like copies of the ground-state PES $V_g(q_1)$ shifted in energy.

The strongly coupled PES for varying numbers of molecules are shown in Fig. 3c,d. We keep the total Rabi frequency constant (corresponding to a scaling of the single-photon field strength with $N^{-1/2}$). Close to equilibrium ($q_1 \approx q_0$), the $N + 1$ surfaces can be clearly classified into a lower and upper polariton PES (light grey), which show significant hybridization with the photonic mode, as well as $N - 1$ 'dark' surfaces (orange) that are almost purely excitonic[19].

As the number of molecules is increased, the local minimum of the lower-polariton PES (the lowest light grey PES) close to the equilibrium position $\mathbf{q}_0$ becomes more and more reminiscent of the pure ground-state PES, making the potential energy barrier to photoisomerization higher and higher. For the cases studied here, the barrier height reaches $\approx 117$ meV for $N = 5$ and $\approx 156$ meV for $N = 50$ molecules, well above the thermal energy at room temperature. This behaviour can be immediately understood from the structure of the polaritonic state: A single excitation is distributed over $N$ molecules and the photonic mode, such that each molecule is in its electronic ground state most of the time. Nuclear motion then takes place mostly on the (stable) ground-state PES, not the (unstable) excited-state PES. This observation resolves the apparent contradiction mentioned above that a single photonic mode would be expected to have less, not more, influence on the internal degrees of freedom when many molecules are involved. Collective strong coupling thus protects the excitation not by directly changing the internal molecular structure, but by creating an excited state with the internal

structure of the uncoupled electronic ground state. This collective protection effect generalizes the 'polaron decoupling' found by Herrera and Spano for molecules with purely harmonic nuclear motion[21] to arbitrary PES. Our PES-based approach furthermore demonstrates that indeed the participation of many molecules is essential for this effect to occur.

In addition to producing a higher reaction barrier, the similarity of the ground and lower polariton PES for large $N$ implies that the Franck-Condon factors, that is, the overlap between nuclear eigenstates in the ground and lower polariton PES, become approximately diagonal. Thus, transitions from the overall ground state to vibrationally excited states in the lower polariton PES become more and more suppressed. In principle, this argument depends on the Condon approximation ($\boldsymbol{\mu}_{LP,g}(\mathbf{q}) \approx \boldsymbol{\mu}_{LP,g}(\mathbf{q}_0)$ over the width of the vibrational ground-state wavepacket). Closer inspection reveals that a similar collective protection effect as for the PES itself exists for the dipole matrix element because most molecules involved in the collective transition are actually at the equilibrium position when following the reaction pathway with motion of only a single molecule. The Condon approximation is thus well-fulfilled for $N \gg 1$ even if the single-molecule dipole matrix element depends strongly on $q$. Photoexcitation then cannot change the vibrational state, such that the excited wavepacket will be close to its vibrational ground state, providing an additional stabilization effect.

Finally, a third effect further improves the stabilization in the lower polariton. Closer inspection of Fig. 3c,d reveals that the lower polariton PES features a narrow avoided crossing (at $q \approx -0.75$ a.u., see inset in Fig. 3d) where it switches from a hybridized collective excited state involving all molecules to essentially the single-molecule excited-state surface. The large wavefunction mismatch makes adiabatic nuclear motion unlikely, and diabatic motion, in which the electronic and photonic degrees of freedom are unchanged, becomes much more likely. This can be shown by constructing diabatic PES close to the avoided crossing, obtained by diagonalizing the coupling between $N - 1$ 'unmoving' molecules and the light mode (giving a very good approximation to the LP PES), which is then coupled to the excited-state PES of the single moving molecule. A short calculation reveals that the transition matrix element between the diabatic LP surface and the single-molecule excited surface is suppressed by a factor $\sim N^{-1/2}$ for a fixed collective Rabi frequency, indicating that the transition to the isomerization surface is indeed strongly suppressed.

## Discussion

The combination of the effects discussed above leads to an almost complete suppression of photoisomerization. The predicted effect

is thus much stronger than the change in the rate of a photoisomerization reaction observed experimentally by Hutchison *et al.*[14]. There are two main reasons for this difference. First, the isomer representing the starting point of the reaction in the experiment was not under strong coupling, such that the initially excited state was not necessarily collective and much less affected by the coupling to the cavity. Second, while we here treat a single confined light mode, the experiments were performed inside a planar microcavity with a continuum of light modes.

To conclude, we have demonstrated the stabilization of excited-state molecular structure and accompanying strong suppression of photochemical reactions under strong coupling of molecules to confined light modes. While already effective in the case of a single coupled molecule, we find that collective coupling of a large number molecules to a single light mode does not actually reduce the influence of strong light-matter coupling on each molecule, but provides even stronger stabilization. This counterintuitive feature can be understood by the additional protection afforded by collective distribution of the excitation over the molecules. These results do not depend on the specifics of the molecular model, such that the observed stabilization is expected to occur for any kind of photochemical reaction that is induced by motion on the excited molecular PES. These results thus pave the way towards a new type of material, created through strong coupling to quantized light modes, for devices such as solar cells.

## Methods
**Molecular model.** We here describe the molecular model in more detail. The adiabatic PES of the bare molecule are constructed in terms of diabatic surfaces $V_A(q)$ and $V_B(q)$ coupled to each other with a coupling $h_0$ that is assumed constant in space. This gives the following electronic Hamiltonian:

$$\hat{H}_{el}(q) = \begin{pmatrix} V_A(q) & h_0 \\ h_0 & V_B(q) \end{pmatrix}. \tag{2}$$

Diagonalization of $\hat{H}_{el}(q)$ returns the ground and excited state PES of Fig. 2a, $V_g(q)$ and $V_e(q)$, together with the adiabatic electronic wavefunctions. This also gives access to the nonadiabatic coupling that controls the transition between ground and excited surfaces at $q \approx 0$, given by $F_{i,j}(q) = \langle i(q)|\partial_q j(q)\rangle$, where $i, j \in \{e, g\}$ and $|i(q)\rangle$ represent the adiabatic electronic states. We note that nonadiabatic transitions in 'real' molecules typically involve conical intersections[31], which only occur in multi-dimensional systems; however, the details of this transition do not influence the results presented.

The complete molecular Hamiltonian is then given by

$$\hat{H}_{mol}(q) = \frac{\hat{P}^2}{2M_q} + \hat{V}(q) + \hat{\Lambda}(q), \tag{3}$$

where $\hat{P}$ is the (diagonal) nuclear momentum operator, $M_q$ is the effective mass for the nuclear coordinate $q$, $\hat{V}(q)$ is the (diagonal) potential operator in the adiabatic basis, and $\hat{\Lambda}(q)$ is the operator of offdiagonal (nonadiabatic) couplings, given by $\hat{\Lambda}(q) = (1/2M_q)(2\hat{F}(q)\partial_q + \hat{G}(q))$, with $G_{i,j}(q) = \partial_q F_{i,j}(q) + F_{i,j}^2(q)$ (ref. 31).

When introducing the coupling to the quantized confined light mode, the total Hamiltonian additionally depends on the dipole moment $\hat{\boldsymbol{\mu}}(q)$, which we set as purely offdiagonal in the adiabatic basis. The ground-excited dipole moment $\boldsymbol{\mu}_{eg}(q)$ typically is approximately constant close to the stable geometries, but changes rapidly close to the nonadiabatic transition due to the sudden polarization effect[32]. We thus choose $|\boldsymbol{\mu}_{eg}(q)| \propto \arctan(q/q_m)$, with $q_m = 0.625$ representing the length scale on which $\boldsymbol{\mu}_{eg}(q)$ changes. As discussed above, the specific shape of $\boldsymbol{\mu}_{eg}(q)$ does not affect the results presented here strongly. This is especially true for $N \gg 1$, where the collective protection effect leads to a dipole matrix element from the ground state to the lower polariton that is almost spatially constant. Diagonalization of the total adiabatic $N$-molecule electron-photon Hamiltonian

$$\hat{H}_{SC} = \omega_c \hat{a}^\dagger \hat{a} + \sum_i \left( \hat{V}(q_i) + \hat{\boldsymbol{\mu}}(q_i) \cdot \mathbf{E}_{1ph}\left(\hat{a}^\dagger + \hat{a}\right) \right) \tag{4}$$

within the single-excitation subspace then yields the strongly coupled (polaritonic) PES. We note that the nonadiabatic couplings in the polaritonic basis are given by new terms $\hat{\Lambda}_{SC}$ due to the basis change, as well as the bare-molecule nonadiabatic couplings $\hat{\Lambda}(q_i)$ transformed to the polaritonic basis.

To evaluate population transfer both in the uncoupled and in the strongly coupled system, we finally solve the time-dependent Schrödinger equation $i\partial_t|\psi(t)\rangle = \hat{H}_{tot}|\psi(t)\rangle$, where $\hat{H}_{tot}$ is the total Hamiltonian without invoking the Born-Oppenheimer approximation, that is, including all nonadiabatic terms. The initial wavefunction is given by direct promotion of the ground-state nuclear wavepacket to the lowest excited state (excited molecular state for no coupling, lower polariton under strong coupling), filtered by the $q$-dependent transition dipole moment from the ground state. This is the initial state that would be obtained after excitation by an ultrashort laser pulse tuned to the excitation energy around the nuclear equilibrium position.

**Data availability.** The data that support the findings of this study are available from the corresponding author on reasonable request.

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

## Acknowledgements

This work has been funded by the European Research Council (ERC-2011-AdG proposal No. 290981), by the European Union Seventh Framework Programme under grant agreement FP7-PEOPLE-2013-CIG-618229, and the Spanish MINECO under contract MAT2014-53432-C5-5-R and the 'María de Maeztu' programme for Units of Excellence in R&D (MDM-2014-0377).

## Author contributions

J.F. and F.J.G.-V. conceived and supervised the work. J.G. performed the numerical calculations. All authors analysed the results and contributed to the writing of the manuscript.
