## [Peer Review File · Nature Communications]

Reviewers' comments:

Reviewer #1 (Remarks to the Author):

This is a fascinating report that presents new research in a topical and rapidly expanding area. The authors provide a theoretical exploration of how a photochemical reaction (in this case photo-isomerization) can be altered through the effect of strong coupling, i.e. the interaction of the molecules involved with a nanophotonic mode. In particular they show a remarkable and unexpected feature - that by coupling a large number of molecules together molecular stabilization can be improved. I think this work will make a valuable addition to the journal and, subject to queries/comments listed below being addressed satisfactorily, I am keen to see the article accepted. Most of my comments are provided in the hope they will improve the clarity of the presentation and make the physics more apparent to readers.

Comments etc. are listed in the same sequence they occur in the manuscript.

1. Page 1, left column, second paragraph (line 30). I suggest changing "and the light" to "and a light".

2. Towards the end of the same paragraph (line 49). I suggest deleting, "to discrete quantized light modes".

3. In the following line (50) I suggest changing, "extremely" to "very".

4. Page 1, right hand column. Here I think the authors need to do more to make their work clearer. They say (lines 67 - 70), "At the top of the barrier, a narrow avoided crossing..... leads to an efficient nonadiabatic transition. First, in the figure I am not able to see the avoided crossing they refer to. Would an inset help? Second, it should be made clear why the presence of this avoided crossing leads to an efficient transition. Perhaps this could be done by contrasting the situation discussed with what would happen if there were no avoided crossing?"

5. Figure 2. Please can the authors be clearer about what the difference is between positive and negative reaction coordinate values. It took me some time to work this out. I think they mean that for negative values the system is in one state and that if the coordinate becomes positive then the system has undergone a transition to the other state (i.e. isomerization has taken place). Can the authors confirm this?

6. Page 3, left hand column. Just below midway down this column the authors say (line 168) the authors say, "The figure clearly shows that the barrier for isomerization...starting from the equilibrium position.....is minimal". I think they are referring to figure 3a (although the panels in the figure are not labelled). I think a cut through these data might help here because it is not clear to me from the data presented that the barrier is minimal - it looks to be a clear barrier to me! It would also help to mark on this figure where the equilibrium position is.

7. Page 3, right hand column. Line 192, "more reminiscent of". It would help if the authors could be more specific, are they referring to the pale coloured curve in the figure?

7. Page 3, right hand column. Line 218. The avoided crossing they mention here is not clear to me. Again, might an insert help?

8. Line 239. I think, "couple" should be "coupled".

Reviewer #2 (Remarks to the Author):

I list my comments below. They are of both methodological and results-oriented character and should all be addressed in detail:

Galego et al. report on suppression of photoisomerization due to strong coupling to the vacuum field of some resonator (photonic or plasmonic). The essence of the work can be summarized as follows: by mixing a photonic (stable) and excitonic (unstable) potential energy surfaces (PES) one can obtain a stable hybrid. This holds irrespectively of the number of molecules N participating in strong coupling and thus is a truly collective effect. The stability of the photonic PES is inherited directly from the stability of the molecular ground state by assuming that its dependence on the reaction coordinate is exactly the same as in the ground state – $V_c(q) = V_g(q) + w_c$.

- This last point is crucial for the whole study, but it is not immediately obvious, at least to me, and thus deserves some further clarification and discussion.
- The height of the barrier in Fig. 2-3 is very low – PES profiles and minima are much shallower than the molecular ground state. The authors should analyze the depth of this barrier with respect to Rabi splitting and thermal energy $k_B T$. If it turns out that the barrier is just a few $k_B T$, I would say that the stability the authors find will not be present in any real situation of finite temperature and presence of phonons. In addition, in any real situation energy of excitation can be greater than exact transition between $|0, g\rangle \rightarrow |-\rangle$, and thus can contain some extra phononic excitations which relax slowly (typically some 1-10 ps) in comparison to the photoisomerization reaction described here (100-200 fs). This all will lead to passage over the barrier and thus stability is lost! With this all findings of this work might be not very useful.
- I am not quite convinced about taking dependence of $\mu_{eg}(q) = \arctan(q/q_m)$. Have the authors tried anything else? Why this specific one?
- The collective strong coupling is undoubtedly very important for practical applications. For this reason, I would expect a much more thorough discussion on the earlier expectations that for sufficiently large N the photoisomerization suppression should disappear, etc. Since the Hamiltonian Eq(1) is straightforward and collective stable behavior seems to naturally follow from that for every molecule making up the ensemble, I wonder where this hypothesis and even fear of that only the single-molecule strong coupling strength should affect chemistry and other processes comes from in the first place in the whole community at large? Discussing this issue will significantly strengthen the paper.
- In the earlier PRX 2015 work by the authors, the subject was about cavity-induced modifications in the ground state. Why this is not present here and if present, how would this affect the stability?
- A minor comment: I would expect a more explicit discussion/referencing of/to Ref. 14, since this work seems to be inspired by that original experiment. The same goes for Ref. 21, which follows similar ideas.

Reviewer #3 (Remarks to the Author):

The manuscript by Galego and coworkers is a very interesting example of how confined electromagnetic modes can nontrivially modify reactivity of molecules which interact with them. The paper shows how one can alter the potential energy surfaces of the molecular systems creating wells and barriers where there were no such at the beginning. A major contribution of this paper is that it addresses the many-molecule case, which has so far been elusive. The authors provide a first step towards an understanding of the many-molecule case which has not been reported in the literature.

The paper reads well and provides substantial physical intuition, and therefore, is likely to attract a

broad and interdisciplinary readership at the intersection of physics, chemistry, and nanophotonics. I enthusiastically support its publication provided the following minor points are addressed:

1. Fig. 2 caption, 5th line: "ground and lower polariton" should be "lower polariton".
2. Line 170: "the barrier for isomerization ... is minimal for motion along only one molecular degree of freedom". The authors should make this statement more formal. In 2D, this is evident from direct calculation as in Fig. 3a, but is this the case in the multidimensional surface? Also, what is the intuition behind this statement?
3. Line 184: Shouldn't it be $N^{0.5}$ rather than $N^{-0.5}$?
4. Paragraph starting in Line 190: This is true only for regions to the left of the barrier. Please make this clear.
5. Paragraph starting in Line 206: This is strictly true only for $\mu(q)=\mu$; however, there is a dependence of μ as a function of q in your model. How is this statement affected by such dependence?
6. Fig. 2: I think the authors should be clearer about why they stop at 260 fs, because the existence of the shallow wells makes one wonder why doesn't the wavepacket keep moving to the right at sufficiently long times. This is specially confusing for (b), where, by eye, the well looks quite shallow.
7. Shall the authors comment how do their results differ from the "polaron decoupling" mechanism advocated by Herrera and Spano in the PRL paper 116, 238301?
8. Shall the authors provide an estimate of N to give the Rabi couplings that they use for their calculations? For that, one might need to have a more parametrized transition dipole moment function. If this is not easy, please give rough estimates.

I. REVIEWER #1

1. Page 1, left column, second paragraph (line 30). I suggest changing "and the light" to "and a light". We did so.
2. Towards the end of the same paragraph (line 49). I suggest deleting, "to discrete quantized light modes". Also done.
3. In the following line (50) I suggest changing, "extremely" to "very". Done.
4. Page 1, right hand column. Here I think the authors need to do more to make their work clearer. They say (lines 67 - 70), "At the top of the barrier, a narrow avoided crossing..... leads to an efficient nonadiabatic transition. First, in the figure I am not able to see the avoided crossing they refer to. Would an inset help? Second, it should be made clear why the presence of this avoided crossing leads to an efficient transition. Perhaps this could be done by contrasting the situation discussed with what would happen if there were no avoided crossing? This is addressed together with the next remark:
5. Figure 2. Please can the authors be clearer about what the difference is between positive and negative reaction coordinate values. It took me some time to work this out. I think they mean that for negative values the system is in one state and that if the coordinate becomes positive then the system has undergone a transition to the other state (i.e. isomerization has taken place). Can the authors confirm this?

We have rephrased the part describing the bare molecule to be clearer, now reading: The ground state PES, $V_g(q)$ (blue line), possesses minima at $q = q_0 \approx -1.05$ a.u. and $q \approx 1$ a.u., corresponding to the stable (e.g., trans-) and metastable (e.g., cis-) isomers, respectively. They are separated by a barrier with a maximum at $q \approx 0$ accompanied by an avoided crossing between the ground and excited state PES, $V_e(q)$ (orange line). In order to ensure a large quantum yield for photoisomerization in the bare molecule, we choose a very narrow avoided crossing (with energy splitting 39 meV, smaller than the width of the lines in Fig. 2). A wavepacket that encounters the avoided crossing then undergoes an efficient nonadiabatic transition (i.e., follows the diabatic surfaces, see Methods), as shown in Fig. 2d using full wavepacket propagation after excitation from the ground to the excited electronic state by an ultrashort laser pulse. For the chosen parameters, the bare model molecule undergoes rapid photoisomerization, with the nuclear wavepacket reaching the second isomer ($q > 0$) within a few hundred fs. In addition, we have added sketches of the trans- and cis-isomers close to the respective minima in Fig. 2a.

6. Page 3, left hand column. Just below midway down this column the authors say (line 168) the authors say, "The figure clearly shows that the barrier for isomerization...starting from the equilibrium position....is minimal". I think they are referring to figure 3a (although the panels in the figure are not labelled). I think a cut through these data might help here because it is not clear to me from the data presented that the barrier is minimal - it looks to be a clear barrier to me! It would also help to mark on this figure where the equilibrium position is. We have changed Fig. 3 to clarify these points. First of all, "minimal" here referred to the fact that the smallest barrier against photoisomerization is found along this direction, not that this barrier is necessary small on an absolute scale. We have thus rewritten the relevant sentence to read: The figure clearly shows that the smallest barrier for isomerization starting from the ground-state equilibrium position $\vec{q}_0 = (q_0, \dots, q_0)$ occurs for motion along only one molecular degree of freedom. In addition, we have modified Fig. 3a to indicate \vec{q}_0 explicitly, as well as indicate the cut shown in Fig. 3(b-d).
7. Page 3, right hand column. Line 192, "more reminiscent of". It would help if the authors could be more specific, are they referring to the pale coloured curve in the figure?

That is correct, we have clarified this in the text: As the number of molecules is increased, the local minimum of the lower-polariton PES (the lowest light gray PES) close to the equilibrium position \vec{q}_0 becomes more and more reminiscent of the pure ground-state PES, making the potential energy barrier to photoisomerization higher and higher.

8. Page 3, right hand column. Line 218. The avoided crossing they mention here is not clear to me. Again, might an insert help?

We have followed this suggestion.

9. Line 239. I think, "couple" should be "coupled".

Correct, and changed.

II. REVIEWER #2

1. The stability of the photonic PES is inherited directly from the stability of the molecular ground state by assuming that its dependence on the reaction coordinate is exactly the same as in the ground state - $V_c(q) = V_g(q) + \omega_c$. This last point is crucial for the whole study, but it is not immediately obvious, at least to me, and thus deserves some further clarification and discussion.

We have now clarified this in the main text: Without light-matter coupling, the photonic excited surface describes the motion of a ground-state molecule with an (uncoupled) photon present in the cavity, and is thus simply a copy of the molecular ground state shifted upwards by the photon energy, $V_c(q) = V_g(q) + \omega_c$ (purple curve in Fig. 2a). It is a direct consequence of the fact that we are here talking about the uncoupled system - i.e., the uncoupled surface $V_c(q)$ just describes the motion of the molecule in its electronic ground state while a photon is present in the cavity, without yet taking into account any coupling between the two.

2. The height of the barrier in Fig. 2-3 is very low - PES profiles and minima are much shallower than the molecular ground state. The authors should analyze the depth of this barrier with respect to Rabi splitting and thermal energy $k_B T$. If it turns out that the barrier is just a few $k_B T$, I would say that the stability the authors find will not be present in any real situation of finite temperature and presence of phonons. In addition, in any real situation energy of excitation can be greater than exact transition between $|0, g\rangle \rightarrow |-\rangle$, and thus can contain some extra phononic excitations which relax slowly (typically some 1-10 ps) in comparison to the photoisomerization reaction described here (100-200 fs). This all will lead to passage over the barrier and thus stability is lost! With this all findings of this work might be not very useful.

We thank the referee for this pertinent comment, as we forgot to mention in the previous version that the barriers we find are significantly higher than the thermal energy, especially for $N \gg 1$. We have now included a discussion of the expected behavior after the initial coherent wavepacket motion discussing this for the single-molecule case: If the coherent excited wavepacket is successfully trapped without undergoing ultrafast isomerization (as in Fig. 2(c,f)), the ultimate fate of the molecule will be determined by two additional effects: On the one hand, the excited wavepacket will thermalize within the excited polariton PES on typical timescales of picoseconds. While the exact values depend on the details of the system, we note that for the model molecule treated here, the barrier height of ≈ 65 meV in Fig. 2 is much larger than the thermal energy $k_B T \approx 26$ meV at room temperature, preventing isomerization. On the other hand, the excited-state wavepacket will simultaneously decay both by radiative and nonradiative processes with timescales typically dominated by the photonic part of the polaritonic PES, ranging from tens of femtoseconds for plasmonic resonances to picoseconds and longer for dielectric structures. We also mention the barrier heights for the many-molecule cases: For the cases studied here, the barrier height reaches ≈ 117 meV for $N = 5$ and ≈ 156 meV for $N = 50$ molecules, well above the thermal energy at room temperature.

In addition, we have clarified that the second effect the referee mentions (phononic excitations accompanying the dipole transition to the polaritonic state) is already included in our calculations: The initial wavepacket on the lower polariton surface in our calculations is started by a sudden transition and thus includes all vibrationally excited states that are reachable from the ground state through a dipole transition. Furthermore, the collective protection encountered in the many-molecule case effectively suppresses these phononic excitations for the case $N \gg 1$ (as discussed in the main text).

3. I am not quite convinced about taking dependence of $\mu_{eg}(q) = \arctan(q/q_m)$. Have the authors tried anything else? Why this specific one?

The reason we chose this form is described in the Methods section: The ground-excited dipole moment $\mu_{eg}(q)$ typically is approximately constant close to the stable geometries, but changes rapidly close to the nonadiabatic

transition due to the sudden polarization effect [32]. There are of course many forms that would reproduce these properties, but it turns out that the specific form does not matter strongly: We have investigated the importance of a spatially dependent $\mu(q)$ following the referee’s comment, as well as a related comment of referee #3. Our findings are discussed in the new version: Thus, transitions from the overall ground state to vibrationally excited states in the lower polariton PES become more and more suppressed. In principle, this argument depends on the Condon approximation ($\mu_{LP,g}(\vec{q}) \approx \mu_{LP,g}(\vec{q}_0)$ over the width of the vibrational ground-state wavepacket). Closer inspection reveals that a similar collective protection effect as for the PES itself exists for the dipole matrix element because most molecules involved in the collective transition are actually at the equilibrium position when following the reaction pathway with motion of only a single molecule. The Condon approximation is thus well-fulfilled for $N \gg 1$ even if the single-molecule dipole matrix element depends strongly on q . Photoexcitation then cannot change the vibrational state, such that the excited wavepacket will be close to its vibrational ground state, providing an additional stabilization effect.

4. The collective strong coupling is undoubtedly very important for practical applications. For this reason, I would expect a much more thorough discussion on the earlier expectations that for sufficiently large N the photoisomerization suppression should disappear, etc. Since the Hamiltonian Eq(1) is straightforward and collective stable behavior seems to naturally follow from that for every molecule making up the ensemble, I wonder where this hypothesis and even fear of that only the single-molecule strong coupling strength should affect chemistry and other processes comes from in the first place in the whole community at large? Discussing this issue will significantly strengthen the paper.

We thank the referee for this comment, which gave us the opportunity to strengthen the analysis and discuss the resolution of the apparent contradiction between our result and the expectation that internal degrees of freedom are only affected by the single-molecule coupling strength (i.e., weakly affected for $N \gg 1$), as shown in several recent papers (Refs. [19,20] in the main text). We have now rewritten the discussion of our findings to explicitly discuss this: This behaviour can be immediately understood from the structure of the polaritonic state: A single excitation is distributed over N molecules and the photonic mode, such that each molecule is in its electronic ground state most of the time. Nuclear motion then takes place mostly on the (stable) ground-state PES, not the (unstable) excited-state PES. This observation resolves the apparent contradiction mentioned above that a single photonic mode would be expected to have less influence on the internal degrees of freedom when many molecules are involved, instead of more. Collective strong coupling thus protects the excitation not by directly changing the internal molecular structure, but by creating an excited state with the internal structure of the uncoupled electronic ground state.

5. In the earlier PRX 2015 work by the authors, the subject was about cavity-induced modifications in the ground state. Why this is not present here and if present, how would this affect the stability?

While our earlier PRX dealt primarily with the (polaritonic) excited states as well, a part of it (section IV) was indeed dedicated to changes in the ground state. We found that these are negligible in currently realistic situations (unless ultrastrong coupling is reached with a *single* molecule). We thus do not explicitly discuss the (negligible) changes in the ground state here.

6. A minor comment: I would expect a more explicit discussion/referencing of/to Ref. 14, since this work seems to be inspired by that original experiment. The same goes for Ref. 21, which follows similar ideas.

We agree that a more explicit discussion of these works could be helpful to better present the context of this work. We have now mentioned Ref. 14 explicitly in the introduction: In particular, Hutchison et al. showed that the rate of a photochemical reaction (photoisomerization from spiropyran to merocyanine) can be modified [14], and discuss the relation to our results explicitly before the conclusions: The combination of the effects discussed above leads to an almost complete suppression of photoisomerization. The predicted effect is thus much stronger than the change in the rate of a photoisomerization reaction observed experimentally by Hutchison *et al.* [14]. There are two main reasons for this difference. First, the isomer representing the starting point of the reaction in the experiment was not under strong coupling, such that the initially excited state was not necessarily collective and much less affected by the coupling to the cavity. Second, while we here treat a single confined light mode, the experiments were performed inside a planar microcavity with a continuum of light modes.

We furthermore added a discussion of Ref. [21] in relation to our work: This collective protection effect generalizes the “polaron decoupling” found by Herrera and Spano for molecules with purely harmonic nuclear motion [21] to arbitrary potential energy surfaces. Our PES-based approach furthermore demonstrates that indeed the participation of *many* molecules is essential for this effect to occur.

III. REVIEWER #3

1. Fig. 2 caption, 5th line: "ground and lower polariton" should be "lower polariton".
Changed.
2. Line 170: "the barrier for isomerization ... is minimal for motion along only one molecular degree of freedom". The authors should make this statement more formal. In 2D, this is evident from direct calculation as in Fig. 3a, but is this the case in the multidimensional surface? Also, what is the intuition behind this statement?
We have expanded on this statement and added the following text: This can be understood from the fact that each of the $N + 1$ uncoupled PES that combine to form the lower polariton describe the nuclear motion of N (for V_C) or $N - 1$ (for $V_E^{(i)}$) molecules in the ground state. The simultaneous motion of more than one molecule thus necessarily corresponds to motion in the ground-state potential wells, i.e., along steep potential barriers. To verify this, we have explicitly checked that the barrier for isomerization rapidly increases with n for simultaneous motion of n molecules ($q_j = q$ for $j \leq n$, and $q_j = q_0$ for $j > n$).
3. Line 184: Shouldn't it be $N^{0.5}$ rather than $N^{-0.5}$?
While the Rabi frequency indeed scales as $N^{1/2}$ for N molecules, we here want to keep the Rabi frequency fixed while N is increased, such that the single-photon electric field strength $E_{1\text{ph}}$ has to scale as $N^{-1/2}$ to make $\Omega_R \propto E_{1\text{ph}}\sqrt{N}$ constant. To clarify that this is a physically meaningful choice, we have added the following footnote when discussing the scaling of Ω_R with N : In this context, it should be noted that since the single-photon electric field strength decreases with the effective mode volume of the EM mode, $E_{1\text{ph}} \propto V^{-1/2}$, the achievable Rabi splitting $\Omega_R \propto \sqrt{N/V}$ depends on the molecular density $\rho \propto N/V$ instead of the absolute number of molecules [30]. This explains why the Rabi splittings achieved in organic systems are within an order of magnitude ($\approx 0.1 - 1$ eV) for EM modes with vastly different mode volumes. This also addresses point 8 raised by this reviewer below.
4. Paragraph starting in Line 190: This is true only for regions to the left of the barrier. Please make this clear.
We have clarified this: As the number of molecules is increased, the local minimum of the lower-polariton PES (the lowest light gray PES) close to the equilibrium position \vec{q}_0 becomes more and more reminiscent of the pure ground-state PES, making the potential energy barrier to photoisomerization higher and higher.
5. Paragraph starting in Line 206: This is strictly true only for $\mu(q)=\mu$; however, there is a dependence of μ as a function of q in your model. How is this statement affected by such dependence?
Please see the answer to question 3 of reviewer #2.
6. Fig. 2: I think the authors should be clearer about why they stop at 260 fs, because the existence of the shallow wells makes one wonder why doesn't the wavepacket keep moving to the right at sufficiently long times. This is specially confusing for (b), where, by eye, the well looks quite shallow.
We have now made clear that case (b) corresponds to an intermediate regime where isomerization is indeed not suppressed, as no barrier is formed yet: At intermediate coupling, where no barrier is formed yet, the reaction is slowed down, but not suppressed (see Fig. 2(b,e)). Once the coupling becomes sufficiently large, a barrier appears and the excited wavepacket is trapped in the local minimum, such that isomerization becomes almost completely suppressed (see Fig. 2(c,f)). For longer times than shown in the figure, decoherence processes should be taken into account (as discussed in the reply to point 2 of reviewer #2).
7. Shall the authors comment how do their results differ from the "polaron decoupling" mechanism advocated by Herrera and Spano in the PRL paper 116, 238301?
We have done so now, please see the reply to question 6 of reviewer #2 for details.
8. Shall the authors provide an estimate of N to give the Rabi couplings that they use for their calculations? For that, one might need to have a more parametrized transition dipole moment function. If this is not easy, please give rough estimates.
Please see the reply to point 3 above.

REVIEWERS' COMMENTS:

Reviewer #1 (Remarks to the Author):

The authors have answer my queries and comments to my satisfaction

Reviewer #2 (Remarks to the Author):

The authors seem to address all my comments. I recommend publication.

A few minor things:

Line 258: ", instead of more". Do the authors mean "instead on one" instead?

Line 307: the word "suppressed" is misspelled.

Reviewer #3 (Remarks to the Author):

The authors have successfully addressed all comments from the referees and I'm happy to support its publication in Nat Comms. congrats!

I. REVIEWER #2

1. Line 258: “, instead of more”. Do the authors mean “instead on one” instead?

This was meant to refer to the strength of the influence of strong coupling, not the number of molecules. We have edited the sentence in line 258 for clarity: **This observation resolves the apparent contradiction mentioned above that a single photonic mode would be expected to have less, not more, influence on the internal degrees of freedom when many molecules are involved.**

2. Line 307: the word “suppressed” is misspelled.

We have corrected the spelling.